# Noise is More Than Just Interference: Information Infusion Networks for Anomaly Detection

## Abstract

3D anomaly detection is a crucial task in computer vision, aiming to identify anomalous points or regions from point cloud data. However, existing methods may encounter challenges when handling point clouds with high intra-class variance, especially for methods that rely on registration techniques. In this study, we propose a novel 3D anomaly detection method, termed Information Gain Block-based Anomaly Detection (IGB-AD), to address the challenges of insufficient anomaly detection information and high intra-class variance. To extract ordered features from 3D point clouds, the technique of Rotation-Invariant Farthest Point Sampling (RIFPS) is first introduced. Then, an Information Perfusion (IP) module composed of stacked Information Gain Blocks (IGB) is proposed to utilize prior noise to provide more distinguishing information for the features, where IGB is designed to utilize noise in a reverse-thinking manner to enhance anomaly detection. Finally, a Packet Downsampling (PD) technique is developed to preserve key information between multiple clusters to solve the complex downsampling situation. The main purpose of the framework is to utilize the effective information within prior noise to provide more detection criteria for anomaly detection. In addition, an Intra-Class Diversity (ICD) 3D dataset is constructed, which contains multiple categories with high class-variance. Experimental results show that the proposed IGB-AD method achieves the State-Of-The-Arts (SOTA) performance on the Anomaly ShapeNet dataset, with an P-AUROC of 81.5% and I-AUROC of 80.9%, and also gains the best performance on the ICD dataset, with an P-AUROC of 57.4% and I-AUROC of 60.2%. Our dataset will be released after acceptance.

## 1 Introduction

3D anomaly detection has emerged as one of the most pivotal topics in computer vision and graphics processing, and extracting distinctive features is crucial for distinguishing between normal and anomalous point clouds. Existing 3D anomaly detection methods can be broadly classified into traditional approaches and deep learning-based methods.

Traditional methods, such as Back to the Feature (BTF), primarily focus on individual 3D structures and use mathematical techniques to design specific point or local descriptive features Horwitz & Hoshen (2022). Additionally, some researchers have attempted to enhance feature descriptors through teacher-student networks Bergmann & Sattlegger (2022). Although these methods have demonstrated promising results, they may face limitations when extracting features with similar structures. Moreover, relying solely on handcrafted features may fail to fully leverage prior knowledge across samples.

Deep learning methods, by incorporating prior knowledge from large datasets, have gradually become a mainstream approach to addressing these limitations. Current deep learning methods can be primarily categorized into embedding-based and reconstruction-based Zhou et al. (2024). The embedding-based methods involve mapping features extracted using transfer learning onto a specific interval for learning. Distributions that do not fall within this interval are classified as anomalies. RegAD Liu et al. (2023) utilizes the PointMAE model trained on large-scale datasets to capture prior information for anomaly detection. However, efficiently extracting useful information from

individual points or feature matrices poses great challenges. When using a pre-trained model for transfer learning, describing each point in a patch is challenging, increasing memory requirements and complexity, while the model's weights may also affect patch descriptions and overall feature extraction accuracy Wang et al. (2023); Zhao et al. (2024). Reconstruction-based methods primarily focus on the network's performance gap between reconstructing normal and anomalous point clouds. To leverage the prior knowledge of transfer learning for improved reconstruction, IMR-Net Li et al. (2023) was proposed, aiming to utilize the capabilities of the pre-trained PointMAE for more detailed reconstructions.

When employing prior transfer learning for feature embedding and point cloud reconstruction, noise plays a crucial role in the data flow. Traditionally, researchers have viewed noise as an element to be removed from the feature matrix, leading to the development of denoising encoders. Vincent et al. (2008). Based on this concept, many anomaly detection networks have been proposed, such as teacher-student distillation networks, which utilize pre-trained teacher models to further mitigate the effects of noise and obtain more accurate 3D anomaly detection features Rudolph et al. (2022). The R3D-AD method attempts to iteratively remove noise using a distillation model, ultimately obtaining the original point cloud Zhou et al. (2024). However, complete noise elimination is often difficult, and an excessive focus on noise reduction can even degrade performance. Building on this, noise may require a more meaningful interpretation in the context of anomaly detection.

Anomaly detection often requires additional information to differentiate between normal and anomalous features, with large language models serving as an effective source of supplementary information Cheng et al. (2024). However, the training cost may present challenges. Noise, as a prior source of information, consists of a combination of various types of data. By removing irrelevant information, the remaining useful data can enhance anomaly detection. This introduces a new perspective on the role of noise in 3D anomaly detection: **To fully harness the valuable information in the prior noise, the key lies in developing a method that separates useful information from the complex noise.** When useful information is extracted, it can be injected into the feature matrix as supplementary data, thereby improving detection accuracy.

To address these challenges, we propose leveraging noise as prior information to improve the generalization of traditional feature descriptors by introducing diversity. Incorporating noise into self-supervised learning expands the inherent feature expressions of normal samples, thereby enhancing anomaly detection and rotational invariance. We introduce the Information Gain Block Anomaly Detection (IGB-AD), which includes an Information Gain Block (IGB) that retains valuable information in the FPFH feature matrix while assigning error ranges to overcome its generalization limitations. The IGB operates in a self-supervised manner, extracting features independent of registration. Additionally, we propose Packet Downsampling (PD), a memory-efficient method for managing diverse point-level features. Our contributions are summarized as follows:

- We propose the IGB module, which uses noise as a prior dependency to extend the range of available information for the feature matrix. Based on IGB, we propose IGB-AD framework. We propose RIFPS to be responsible for the ordered initial feature matrix extraction. Furthermore, we propose an IP module based on IGB to inject information into the initial feature matrix. Finally, PD is used to perform better downsampling.

- We introduce the ICD dataset, the first 3D anomaly detection dataset with multiple sub-classes, offering new possibilities and challenges for 3D anomaly detection.

- Our IGB-AD framework leverages noise to overcome the limitations of traditional descriptors, compensating for insufficient information in the feature matrix and alleviating the challenges posed by high intra-class variance. It achieves State-Of-The-Arts (SOTA) performance on the Anomaly-ShapeNet dataset with I-AUROC 80.9% and P-AUROC 81.5%, and on our custom ICD dataset, it achieves I-AUROC 60.2% and P-AUROC 57.4%, significantly improving anomaly detection.

## 2 RELATED WORK

3D anomaly detection for point clouds is crucial in applications like autonomous vehicle navigation and industrial inspection. Deep learning approaches have gained prominence, using neural

networks to learn complex point cloud structures. Methods like embedding-based Approach and reconstruction-based approach are key in anomaly detection.

## 2.1 EMBEDDING-BASED APPROACH

In order to obtain features that can distinguish anomalies, a embedding-based method extracts plaque or point-level features from normal samples and stores them in the memory bank. In the inference stage, extracted features of test samples and compared them with those in the memory bank to obtain anomalies. Embedding-based methods are often based on traditional methods of local feature extraction directly or feature extraction methods from transfer learning.

Traditional methods for point cloud feature extraction, such as Fast Point Feature Histograms (FPFH) Szalai-Gindl & Varga (2024), have played a significant role in providing essential feature descriptors for point clouds. While these approaches are effective in capturing local geometric properties, the limited ability to transfer learned features between different point clouds undermines their effectiveness, especially in tasks requiring high generalization, such as anomaly detection.

In contrast, embedding-based methods, grounded in deep learning, aim to address these limitations by focusing on learning distinctive representations of normal point clouds directly from raw data. These methods prioritize efficient feature representation in memory banks for anomaly detection and leverage pre-trained models, such as PointNet++, PointMLP, and Point Transformer Qi et al. (2017); Ma et al. (2022); Zhao et al. (2021), which significantly enhance the feature extraction process. PointMAE Pang et al. (2022) further improves point cloud understanding through masked autoencoders, offering more refined representations. Furthermore, contrastive learning has been proposed as a method to enhance the representation in memory banks Zhu et al. (2024). By incorporating this technique, the ability to distinguish between normal and anomalous point clouds is substantially improved, providing a more robust solution for anomaly detection tasks. This shift from traditional methods to deep learning-based approaches represents a significant advancement in achieving both efficient and generalizable feature representations in point cloud analysis. However, deep learning methods often have large memory consumption, inaccurate description at the point or patch level, and weak differentiation, and still need to be further improved.

## 2.2 RECONSTRUCTION-BASED APPROACH

Reconstruction-based methods detect anomalies by focusing on the reconstruction error between normal and anomalous point clouds. The surface of point clouds is characterized by indirect and direct representation at different feature levels.

Initially, emphasis was placed on reconstructing point clouds in representational space, and accurate original point cloud reconstruction was often not the goal. For instance, methods based on predicting signed distance functions (SDF) were used to indirectly reconstruct surfaces, offering a way to approximate point cloud geometry without requiring exact reconstructions Chu et al. (2023).

More recently, the focus has shifted toward achieving more accurate reconstructions. IMRNet, an extension of PointMAE, reconstructs anomalous clouds into their normal counterparts and detects anomalies by comparing the differences between the original and reconstructed models Li et al. (2023). Due to the thermal diffusion process in the evolving thermodynamic and kinetic system, the researchers proposed a more accurate R3D-AD reconstruction method, which can be more focused on the more subtle reconstruction process of the 3D surface. Zhou et al. (2024). Despite these advances, achieving precise reconstruction remains challenging, as difficulties in accurately capturing complex details often result in suboptimal detection performance. Researchers need a more accurate characterization of normal and abnormal structures.

## 3 INFORMATION GAIN BLOCK-BASED ANOMALY DETECTION

The key challenge in 3D anomaly detection is to develop an effective method for self-supervised identification of those points that deviate from the normal pattern when only normal samples are available. Therefore, this paper proposes an Information Gain Block-Based Anomaly Detection (IGB-AD) framework, which consists of three parts: (1) Rotation-Invariance Farthest Point Sampling (RIFPS), (2) Information Perfusion (IP) based on Information Gain Block (IGB), and (3)

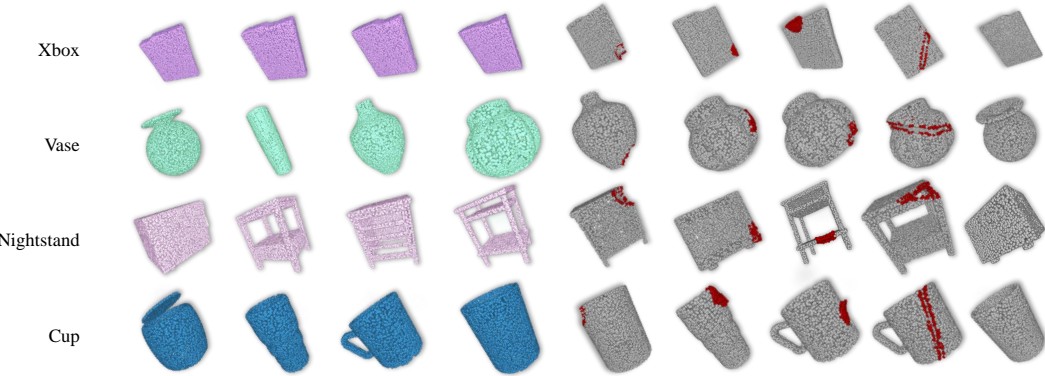

Figure 1: **Visualization of our ICD datasets**. We show a training set for some of the classes, along with a sample for each exception. The training set for each class consists of four objects of the same species with different morphologies and produces associated exceptions. The complete data set is presented in the appendix A.

Packet Downsampling (PD). The overall framework of the proposed method is illustrated in Figure 2. The pseudo-code is shown in the Appendix D.

## 3.1 ROTATION-INVARIANT FARTHEST POINT SAMPLING

To diminish the reliance on registration and enhance the effective utilization of noise within the IGB module, we propose Rotation-Invariant Farthest Point Sampling (RIFPS) to ensure rotation-invariant feature extraction of point clouds. We first calculate the geometric center and find the farthest point:

$$\mathbf{C} = \frac{1}{N} \sum_{i=1}^{N} \mathbf{p}_i, \quad D_i = \|\mathbf{p}_i - \mathbf{C}\|_2. \tag{1}$$

where $\mathbf{C}$ is the geometric center, $\mathbf{p}_i$ the $i$-th point, $N$ the number of points, and $D_i$ the Euclidean distance from $\mathbf{p}_i$ to $\mathbf{C}$. The point $\mathbf{p}_f$ maximizing $D_i$ is selected as the farthest point.

Using $\mathbf{p}_f$ as a reference, we apply Farthest Point Sampling (FPS) and compute the FPFH feature matrix:

$$\mathbf{FPFH}_{\text{inv}} = \text{FPFH}(\mathbf{p}_f, \mathbf{p}_1, \ldots, \mathbf{p}_n). \tag{2}$$

where $\mathbf{FPFH}_{\text{inv}}$ is the FPFH matrix with constant order, and $\mathbf{p}_1, \ldots, \mathbf{p}_n$ are FPS-sampled points. This ensures that feature extraction is sequenced consistently across different point cloud directions, making it robust for anomaly detection.

## 3.2 INFORMATION GAIN BLOCK

A critical aspect of anomaly detection lies in acquiring sufficient distinctive information to effectively differentiate between normal and abnormal instances. Therefore, we use noise as prior information to enhance the information and propose an Information Gain Block (IGB) for better anomaly detection. Extracting useful information from the noise is the task of IGB, and the IGB module gradually transforms noise into useful information by eliminating irrelevant parts. In order to better understand the IGB process, we give a further explanation in Appendix C.

Inspired by the *Central Limit Theorem (CLT)*, we extract information from Gaussian noise to enhance feature diversity. According to the *CLT*, Gaussian noise $Z$ can be decomposed into useful gain information $X$ and irrelevant noise $Y$:

$$Z = X + Y. \tag{3}$$

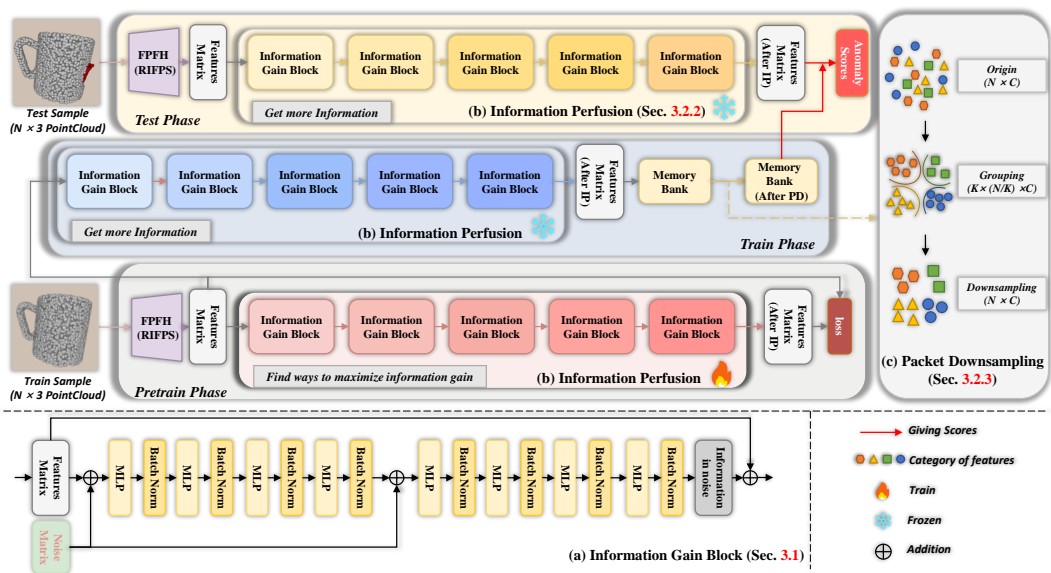

Figure 2: An overview of our approach. In the figure, we represent schematics of (a) IGB, (b) IP, and (c) PD. In the pre-training stage, we use normal sample ordered feature matrix to train IP. In the training stage, we enhanced the features of the ordered samples through the frozen IP layer, and then stored them in the memory bank for further processing by PD. In the test phase, after enhancing the ordered samples by using the frozen IP layer, anomaly detection is carried out by comparing the feature matrix with the features in the memory library.

where $Y$ is modeled as irrelevant noise, $X$ represents the target information. According to the *CLT*, $Y$ approximates a normal distribution, making $Z$ Gaussian. The MLP extracts $X$ from $Z$ as follows:

$$Z \to X : f_{\text{MLP}}(Z, F) = X. \tag{4}$$

where $Z$ is Gaussian noise, $F$ is the reference feature, and $X$ is the extracted information. The MLP learns to extract $X$ by minimizing irrelevant noise $Y$.

We further formalize this process as *Maximum Likelihood Estimation (MLE)*. By maximizing the likelihood function $p(Z \mid X)$, we ensure that the extracted $X$ contains the most relevant information:

$$\hat{X}_{\text{MLE}} = \arg\max_X p(Z \mid X). \tag{5}$$

This reinforces the extraction process, focusing on the most useful parts of $Z$. We use four MLPs to reduce the dimensionality, removing excess noise, and then increase the dimension to obtain effective information. The process is expressed as:

$$Z \to X : IGB(Z, F) = X. \tag{6}$$

This transforms Gaussian noise $Z$ into useful information $X$, guided by the feature $F$. To ensure $X$ maintains the semantic range of $F$, we propose the following loss function:

$$
\begin{aligned}
L_{\text{total}} &= \beta \cdot L_{\text{smooth\_L1}} + \lambda \cdot (1 - L_{\text{richness}}), \\
L_{\text{smooth\_L1}} &= \text{SmoothL1Loss}(F, F + X), \\
L_{\text{richness}} &= \text{sigmoid}(\alpha \cdot \text{mean}(\text{Var}(F + X))).
\end{aligned}
\tag{7}
$$

The model emphasizes both preserving the original information and extending more information. The Smooth L1 Loss preserves essential information, while the Richness Loss encourages feature

variability. This design balances information preservation and representation enhancement, with MLE maximizing the likelihood of extracting relevant information.

### 3.3 INFORMATION PERFUSION

In order to get more refined and rich features, we try to use more stacks for continuous perfusion. We used IGB to construct an Information Perfusion (IP) process, composed of multiple IGB stacks. IP infuses the FPFH feature matrix with prior information from noise via multi-layer IGB. The IP process is formulated as:

$$\mathbf{F}^{(i)} = \mathbf{F}^{(i-1)} + \text{IGB}_i(\mathbf{Z}, \mathbf{F}^{(i-1)}), \quad i = 1, 2, \ldots, k. \tag{8}$$

where $\mathbf{F}^{(0)}$ is the initial FPFH feature matrix, $\mathbf{Z}$ is the noise input, and $\text{IGB}_i$ is the $i$-th IGB module. The output $\mathbf{F}^{(i)}$ is obtained by adding the output of the previous layer to the current IGB module, iterating from $i = 1$ to layer $k$, yielding the final enhanced feature matrix $\mathbf{F}^{(k)}$. Each IGB layer enhances $\mathbf{F}$ by adding gain information from noise $\mathbf{Z}$, iteratively enriching its representation for anomaly detection.

### 3.4 PACKET DOWNSAMPLING

In scenarios involving multiple subclasses, mutual interference among the subclasses often plays a critical role. To select features that are equally robust in both multi-subclass and conventional contexts, and to optimize memory repositories with large-scale point-level features, we propose a Mahalanobis distance-based greedy core-set clustering selection method. This approach aims to select representative samples while maximizing both intra-class and inter-class feature diversity.

K-Means clustering divides the feature bank into $K$ clusters, followed by Mahalanobis distance to select representative samples. The clustering and distance are defined as:

$$\min_C \sum_{i=1}^{N} \left( (x_i - \mu_c)^\top \Sigma^{-1} (x_i - \mu_c) \right),$$

$$\Sigma = \frac{1}{N-1} \sum_{i=1}^{N} (x_i - \mu)(x_i - \mu)^\top, \quad \Sigma^{-1} = \text{Inverse}(\Sigma). \tag{9}$$

where $x_i$ is the $i$-th data point, $\mu_c$ the centroid of cluster $c$, and $\Sigma$ the covariance matrix. Mahalanobis distance accounts for the covariance structure by incorporating $\Sigma$ and its inverse $\Sigma^{-1}$, normalizing variance along each feature dimension.

A greedy algorithm selects samples maximizing Mahalanobis distance from previous selections. To handle density variations, k-nearest neighbors (k-NN) adjusts clustering parameters $\varepsilon$ and min_samples:

$$\varepsilon_i = \frac{1}{k} \sum_{j=1}^{k} d(x_i, x_j),$$

where $d(x_i, x_j)$ is the distance between $x_i$ and its $j$-th nearest neighbor. This computes the average distance to the $k$-nearest neighbors, yielding an adaptive $\varepsilon$ for density variations. The min_samples is set to $k \times 2$ for robust density estimation.

### 3.5 ANOMALY SCORE CALCULATION

We adopt a PatchCore-like scoring approach Roth et al. (2022), constructing a memory bank $\mathcal{M}$ during training with features $F$. In inference, new features $f_{\text{test}}$ are compared to the memory bank to compute point-wise scores, as expressed mathematically:

$$s(f_{\text{test}}) = \min_{m \in \mathcal{M}} |f_{\text{test}} - m|_2, \tag{10}$$

where $s(f_{\text{test}})$ denotes the anomaly score of the feature $f_{\text{test}}$, and $|f_{\text{test}} - m|2$ represents the Euclidean distance between the feature $f_{\text{test}}$ and a feature $m$ from the memory bank. After obtaining the scores for each feature, we perform normalization, which can be expressed as:

$$s_{\text{norm}}(f_{\text{test}}) = \frac{s(f_{\text{test}}) - \min(s)}{\max(s) - \min(s)}, \tag{11}$$

Where $s_{\text{norm}}(f_{\text{test}})$ is the normalized score, and $\min(s)$ and $\max(s)$ are the minimum and maximum scores. This method efficiently normalizes anomaly scores.

## 4 EXPERIMENTS

In this section, we first introduce the previous data sets and our proposed Intra-Class Diversity (ICD) datasets. We then tested IGB-AD on the described data set and performed ablation experiments. Both experimental results and ablation experiments validate the rationality and overall effectiveness of each part of our IGB-AD framework.

### 4.1 DATASETS

Comparative experiments were performed on two mainstream datasets: Intra-Class Diversity and Anomaly-ShapeNet datasets.

**Anomaly-ShapeNet** comprises 40 categories, with 1,600+ positive and negative samples. Each category's training set contains 4 normal samples, while test sets include both normal and anomalous samples exhibiting various defects.

**Ours: Intra-Class Diversity (ICD)** datasets introduce new challenges for anomaly detection, the training set consists of four morphologically distinct subspecies per class, and the test set is derived from each of these subspecies. This poses a unique challenge for models, requiring them to effectively extract features from individual samples while simultaneously capturing the variations across different subspecies. The dataset consists of four subspecies, with the test set containing between 41 and 64 samples per class, covering both normal and anomalous variations of each subspecies. A partial visualization is shown in Figure 1, and a full description is presented in the appendix A.

### 4.2 IMPLEMENTATION

**Baselines.** We selected BTF Horwitz & Hoshen (2022), M3DM Wang et al. (2023), PatchCore Roth et al. (2022), CPMF Cao et al. (2023), RegAD Liu et al. (2023), R3D-AD Zhou et al. (2024) and IMRNet Li et al. (2023) for comparison. Note that BTF(FPFH) denotes that we incorporate fast point feature histogram. The results of these methods are obtained through publicly available code or referenced papers.

**Evaluation Metrics.** For the anomaly detection task, we use P-AUROC ($\uparrow$) to evaluate pixel-level anomaly localization capability and I-AUROC ($\uparrow$) to evaluate object-level anomaly detection capability. Higher values for both metrics indicate a more robust anomaly detection capability.

**Experimental Details.** The experiments were conducted on a server equipped with an RTX 3090 (24GB) GPU and a 14 vCPU Intel(R) Xeon(R) Gold 6330 CPU @ 2.00GHz. We employed the AdamW optimizer, with the pre-training phase set to 50 epochs, an initial learning rate of 0.001, and cosine annealing reducing the learning rate to 0.000001. Throughout the experiments, the number of IGB layers was fixed at 5, which represents a moderate configuration. The Settings for the other comparison models use the Settings in their paper or in the published method.

### 4.3 RESULTS

**Comparisons on ICD.** We quantitatively analyze the Image-level anomaly detection results in Table1. Our method shows superiority in that we assign a precise score to each point, which yields

| | I-AUROC | | | | | | | | | | |
|---|---|---|---|---|---|---|---|---|---|---|---|
| Method | bottle | cup | desk | door | keyboard | night_stand | radio | vase | xbox | cone | Mean |
| BTF (Raw) | 0.017 | 0.073 | 0.006 | 0.006 | 0.006 | 0.011 | 0.023 | 0.125 | 0.006 | 0.030 | 0.030 |
| BTF (FPFH) | 0.291 | 0.412 | 0.597 | 0.504 | 0.418 | 0.415 | 0.340 | 0.430 | 0.436 | 0.587 | 0.443 |
| M3DM | 0.118 | 0.203 | 0.206 | 0.098 | 0.157 | 0.161 | 0.428 | 0.237 | 0.039 | 0.012 | 0.166 |
| Patchcore (FPFH) | 0.667 | 0.548 | 0.550 | 0.647 | 0.539 | 0.479 | 0.591 | 0.594 | 0.608 | 0.591 | 0.581 |
| Patchcore (PointMAE) | 0.427 | 0.643 | 0.517 | 0.010 | 0.101 | 0.452 | 0.435 | 0.486 | 0.314 | 0.578 | 0.396 |
| RegAD | 0.362 | 0.496 | 0.321 | 0.005 | 0.065 | 0.477 | 0.314 | 0.532 | 0.253 | 0.503 | 0.333 |
| Ours | 0.684 | 0.526 | 0.599 | 0.640 | 0.598 | 0.515 | 0.612 | 0.648 | 0.601 | 0.585 | 0.602 |

Table 1: The experimental results I-AUROC (↑) for anomaly detection across 10 categories of ICD. The best and the second-best results are highlighted in **red** and **blue**, respectively. Our model achieved the best average performance across the 10 categories for I-AUROC. The results of P-AUROC are presented in the appendix B

| | I-AUROC | | | | | | | | | | | | | |
|---|---|---|---|---|---|---|---|---|---|---|---|---|---|---|
| Method | cap0 | cap3 | helmet3 | cup0 | bowl4 | vase3 | headset1 | eraser0 | vase8 | cap4 | vase2 | vase4 | helmet0 | bucket1 |
| BTF (Raw) | 0.668 | 0.527 | 0.526 | 0.403 | 0.664 | 0.717 | 0.515 | 0.525 | 0.424 | 0.648 | 0.410 | 0.425 | 0.553 | 0.321 |
| BTF (FPFH) | 0.618 | 0.522 | 0.444 | 0.586 | 0.609 | 0.699 | 0.490 | 0.719 | 0.668 | 0.520 | 0.546 | 0.510 | 0.571 | 0.633 |
| M3DM | 0.557 | 0.423 | 0.374 | 0.539 | 0.464 | 0.439 | 0.617 | 0.627 | 0.663 | 0.777 | 0.737 | 0.476 | 0.526 | 0.501 |
| Patchcore (FPFH) | 0.580 | 0.453 | 0.404 | 0.600 | 0.494 | 0.449 | 0.637 | 0.657 | 0.662 | 0.757 | 0.721 | 0.506 | 0.546 | 0.551 |
| Patchcore (PointMAE) | 0.589 | 0.476 | 0.424 | 0.610 | 0.501 | 0.460 | 0.627 | 0.677 | 0.663 | 0.727 | 0.741 | 0.516 | 0.556 | 0.561 |
| CPMF | 0.601 | 0.551 | 0.420 | 0.497 | 0.683 | 0.582 | 0.458 | 0.689 | 0.529 | 0.553 | 0.582 | 0.514 | 0.555 | 0.601 |
| RegAD | 0.693 | 0.725 | 0.367 | 0.510 | 0.663 | 0.650 | 0.610 | 0.343 | 0.620 | 0.643 | 0.605 | 0.500 | 0.600 | 0.752 |
| IMRNet | 0.737 | 0.775 | 0.573 | 0.643 | 0.676 | 0.700 | 0.676 | 0.548 | 0.630 | 0.652 | 0.614 | 0.524 | 0.597 | 0.771 |
| R3D-AD | 0.822 | 0.730 | 0.707 | 0.822 | 0.744 | 0.742 | 0.795 | 0.890 | 0.721 | 0.681 | 0.752 | 0.630 | 0.757 | 0.756 |
| Ours | 0.933 | 0.835 | 0.491 | 1.000 | 0.982 | 0.827 | 0.729 | 0.948 | 0.939 | 0.777 | 0.824 | 0.603 | 0.725 | 0.651 |

| Method | bottle3 | vase0 | bottle0 | tap1 | bowl0 | bucket0 | vase5 | vase1 | vase9 | ashtray0 | bottle1 | tap0 | phone | cup1 |
|---|---|---|---|---|---|---|---|---|---|---|---|---|---|---|
| BTF (Raw) | 0.568 | 0.531 | 0.597 | 0.573 | 0.564 | 0.617 | 0.585 | 0.549 | 0.564 | 0.578 | 0.510 | 0.525 | 0.563 | 0.521 |
| BTF (FPFH) | 0.322 | 0.342 | 0.344 | 0.546 | 0.509 | 0.401 | 0.409 | 0.219 | 0.268 | 0.420 | 0.546 | 0.560 | 0.571 | 0.610 |
| M3DM | 0.510 | 0.423 | 0.574 | 0.739 | 0.634 | 0.309 | 0.317 | 0.427 | 0.663 | 0.577 | 0.637 | 0.754 | 0.357 | 0.556 |
| Patchcore (FPFH) | 0.572 | 0.455 | 0.604 | 0.766 | 0.504 | 0.469 | 0.417 | 0.423 | 0.660 | 0.587 | 0.667 | 0.753 | 0.388 | 0.586 |
| Patchcore (PointMAE) | 0.650 | 0.447 | 0.513 | 0.538 | 0.523 | 0.593 | 0.579 | 0.552 | 0.629 | 0.591 | 0.601 | 0.458 | 0.488 | 0.556 |
| CPMF | 0.405 | 0.451 | 0.520 | 0.697 | 0.783 | 0.482 | 0.618 | 0.345 | 0.609 | 0.353 | 0.482 | 0.359 | 0.509 | 0.499 |
| RegAD | 0.525 | 0.533 | 0.486 | 0.641 | 0.671 | 0.610 | 0.520 | 0.702 | 0.594 | 0.597 | 0.695 | 0.676 | 0.414 | 0.538 |
| IMRNet | 0.640 | 0.533 | 0.552 | 0.696 | 0.681 | 0.580 | 0.676 | 0.757 | 0.594 | 0.671 | 0.700 | 0.676 | 0.755 | 0.757 |
| R3D-AD | 0.781 | 0.788 | 0.733 | 0.900 | 0.819 | 0.683 | 0.757 | 0.729 | 0.718 | 0.833 | 0.737 | 0.736 | 0.762 | 0.757 |
| Ours | 0.991 | 0.800 | 0.895 | 0.696 | 0.978 | 0.924 | 0.552 | 0.795 | 0.618 | 0.891 | 0.916 | 0.709 | 0.995 | 0.721 |

| Method | vase7 | helmet2 | cap5 | shelf0 | bowl5 | bowl3 | helmet1 | bowl1 | headset0 | bag0 | bowl2 | jar | Mean |
|---|---|---|---|---|---|---|---|---|---|---|---|---|---|
| BTF (Raw) | 0.448 | 0.602 | 0.373 | 0.164 | 0.417 | 0.385 | 0.349 | 0.264 | 0.378 | 0.410 | 0.525 | 0.420 | 0.493 |
| BTF (FPFH) | 0.518 | 0.542 | 0.586 | 0.609 | 0.699 | 0.490 | 0.719 | 0.668 | 0.520 | 0.546 | 0.510 | 0.424 | 0.528 |
| M3DM | 0.657 | 0.623 | 0.639 | 0.564 | 0.409 | 0.617 | 0.427 | 0.663 | 0.577 | 0.537 | 0.684 | 0.441 | 0.552 |
| Patchcore (FPFH) | 0.693 | 0.425 | 0.790 | 0.494 | 0.558 | 0.537 | 0.484 | 0.639 | 0.583 | 0.571 | 0.615 | 0.472 | 0.568 |
| Patchcore (PointMAE) | 0.650 | 0.447 | 0.538 | 0.523 | 0.593 | 0.579 | 0.552 | 0.629 | 0.591 | 0.601 | 0.458 | 0.483 | 0.562 |
| CPMF | 0.397 | 0.462 | 0.697 | 0.685 | 0.685 | 0.658 | 0.589 | 0.639 | 0.643 | 0.643 | 0.625 | 0.610 | 0.559 |
| RegAD | 0.462 | 0.614 | 0.467 | 0.688 | 0.593 | 0.348 | 0.381 | 0.525 | 0.537 | 0.706 | 0.490 | 0.592 | 0.572 |
| IMRNet | 0.635 | 0.641 | 0.652 | 0.603 | 0.710 | 0.599 | 0.600 | 0.702 | 0.720 | 0.660 | 0.685 | 0.780 | 0.661 |
| R3D-AD | 0.771 | 0.633 | 0.670 | 0.696 | 0.656 | 0.767 | 0.720 | 0.778 | 0.738 | 0.720 | 0.741 | 0.838 | 0.749 |
| Ours | 0.767 | 0.887 | 0.730 | 0.852 | 0.526 | 0.870 | 0.600 | 0.878 | 0.840 | 0.710 | 1.000 | 0.952 | 0.809 |

Table 2: The experimental results I-AUROC (↑) for anomaly detection across 40 categories of Anomaly-ShapeNet. The best and the second-best results are highlighted in **red** and **blue**, respectively. Our model achieved the best average performance across the 40 categories for both metrics. Our approach also achieves the best performance in the I-AUROC, due to the length shown in the appendix B.

60.2% I-AUROC, outperforming previous methods. The diversity of subspecies in this dataset challenges the accurate classification of samples. Previous methods, especially those based on registration, have encountered great challenges on this data set

**Comparisons on Anomaly-ShapeNet.** We quantitatively analyze the pixel-level anomaly detection results in Table 5, and Image-level anomaly detection results in Table 2. Our method shows great superiority in that we assign a precise score to each point, which exhibits 81.5% P-AUROC and yields 80.9% I-AUROC, outperforming previous methods.

## 4.4 ABLATION STUDY

We conducted ablation experiments on the components of the IGB-AD framework using the ICD dataset, with the results shown in Table 3. We evaluated the impact of the number of IGB layers and block downsampling (PD) on the model's performance. For the challenging ICD dataset, increasing the number of IGB layers significantly enhanced the anomaly detection capability. Increasing

| Module | | | ICD | | | | |
|---|---|---|---|---|---|---|---|
| PD | IP | IGB | I-AUROC | P-AUROC | I-AUPRO | P-AURPO | Time Cost |
| ✓ | ✓ | 5 | **0.6024** | 0.5741 | **0.6265** | 0.0190 | 0.2492 |
| ✓ | ✓ | 4 | 0.5963 | 0.5822 | 0.6143 | 0.0193 | 0.2285 |
| ✓ | ✓ | 3 | 0.5902 | 0.5836 | 0.6075 | 0.0190 | 0.2256 |
| ✓ | ✓ | 2 | 0.5815 | 0.5782 | 0.6006 | 0.0189 | 0.1991 |
| ✓ | ✓ | 1 | 0.5794 | **0.5898** | 0.6042 | **0.0196** | 0.1849 |
| | ✓ | 5 | 0.5986 | 0.5753 | 0.6108 | 0.0187 | 0.2229 |
| ✓ | | \ | 0.5842 | 0.5816 | 0.6002 | 0.0192 | 0.1726 |

Table 3: Ablation results on ICD datasets. IGB Indicates the number of layers in use. If the IP layer is not used, IGB is not supported.

from one layer to five layers, the I-AUROC improved from 0.5794 to 0.6024, but the inference time also increased accordingly. This is because more IGB layers can more fully learn prior information, thereby more effectively enhancing the useful information within the noise. However, too many IGB layers may lead to excessive computational cost, reducing the model's efficiency. Block downsampling (PD) further improved the model's performance by selecting more representative samples. For example, when using 5 IGB layers, adding PD increased the accuracy from 0.5986 to 0.6024. This indicates that PD plays a critical role in optimizing sample selection and enhancing the model's generalization ability. Notably, when using only the complete IP layer without PD, the model's performance remains strong, outperforming the cases of using only PD and 4 IGB layers, as well as using only PD. This suggests that while PD makes an important contribution to performance improvement, IGB plays a more critical role in the model, primarily enhancing the model's anomaly detection capability by optimizing sample selection.

The selection of Gaussian distribution as the noise model is based on its independent randomness. Under the *independent identically distributed (IID)* noise hypothesis, each sample is independent of the others and follows the same probability distribution. This independence ensures that the noise does not exhibit spatio-temporal dependence, allowing each noise sample to be modeled individually. This not only simplifies the calculation, but also meets our expectation that the added information for each eigenvalue ensures that the sum of multiple independent noise sources will approximate a normal distribution by *CLT*. Thus, Gaussian noise effectively captures uncertainties in independent random processes, making it a robust and computationally tractable choice for noise modeling. In contrast, other types of noise, such as autoregressive or Poisson noise, introduce dependencies or discreteness and lack the simplicity and universality of Gaussian noise.

## 5 CONCLUSION

To address the challenges of insufficient information for anomaly detection and high intra-class variance, we propose the Information Gain Block Anomaly Detection (IGB-AD) framework. The framework first proposes Rotation-Invariant Farthest Point Sam- pling (RIFPS) to ensure the order of the feature matrix, and then proposes an Information Perfusion (IP) composed of multiple Information Gain Block (IGB). The IGB layer is a module that can learn noise as prior information. Finally, Packet Downsampling (PD) is proposed to further reduce the influence of intra-class variance. In order to further verify the ability of the model to face large cases of intra-class variance, we also propose an Intra-Class Diversity dataset (ICD). Experimental results demonstrate that the proposed IGB-AD method achieves State-Of-The-Arts (SOTA) performance on the Anomaly ShapeNet dataset, with P-AUROC of 81.5% and I-AUROC of 80.9%, and outperforms on the ICD dataset, with P-AUROC of 57.4% and I-AUROC of 60.2%. We incorporate noise as prior information into the features through a self-supervised approach to obtain more informative and discriminative representations. This presents a novel avenue for addressing the challenges of insufficient information and high intra-class variance in anomaly detection. **Limitations:** Given the inherent randomness and complexity of noise, quantitatively assessing the precise amount of information injected into the feature matrix remains challenging. Establishing a mechanism for controlled information injection will thus be a key objective in our future research endeavors.

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

## A   3D DATASET: INTRA-CLASS DIVERSITY

| Datasets | Year | Format | Class | Subspecies | Point Range | Anomaly Types |
|---|---|---|---|---|---|---|
| MVTec3D-AD | 2021 | RGB/D | 10 | \ | 10K-30K | 3~5 |
| Eyecandies | 2022 | RGB/D/N | 10 | \ | \ | 3 |
| Real3D-AD | 2023 | PointCloud | 12 | \ | 35K-780K | 2 |
| Anomaly-ShapeNet | 2023 | PointCloud | 40 | \ | 8K-30K | 6 |
| ModelNet-AD (Ours) | 2024 | PointCloud | 10 | 4 | 45K-55K | 4 |

Table 4: Comparison between the proposed ICD and existing mainstream 3D anomaly detection datasets.

Our dataset introduces new challenges for anomaly detection: the training set consists of four morphologically distinct subspecies per class, and the test set is derived from each of these subspecies. This poses a unique challenge for models, requiring them to effectively extract features from individual samples while simultaneously capturing the variations across different subspecies. A visualization of our dataset is provided in Figure 3.

### A.1   PREVIOUS WORK

The emergence of some data sets provides an experimental basis for 3D anomaly detection as shown in Table 4. For example, the earliest MVtec and Eyecandies3D-AD datasets were used for 3D anomaly detection using 2.5D point clouds Bergmann et al. (2022); Bonfiglioli et al. (2022). On this basis, by scanning the real point cloud, the researchers propose the Real3D-AD dataset, which is a high-precision scanning dataset and provides the possibility for entity anomaly detection Liu et al. (2023). Then came the Anomala-Shapenet dataset, a dataset synthesized on ShapeNet that provides 40 cases for Anomaly detection Li et al. (2023).

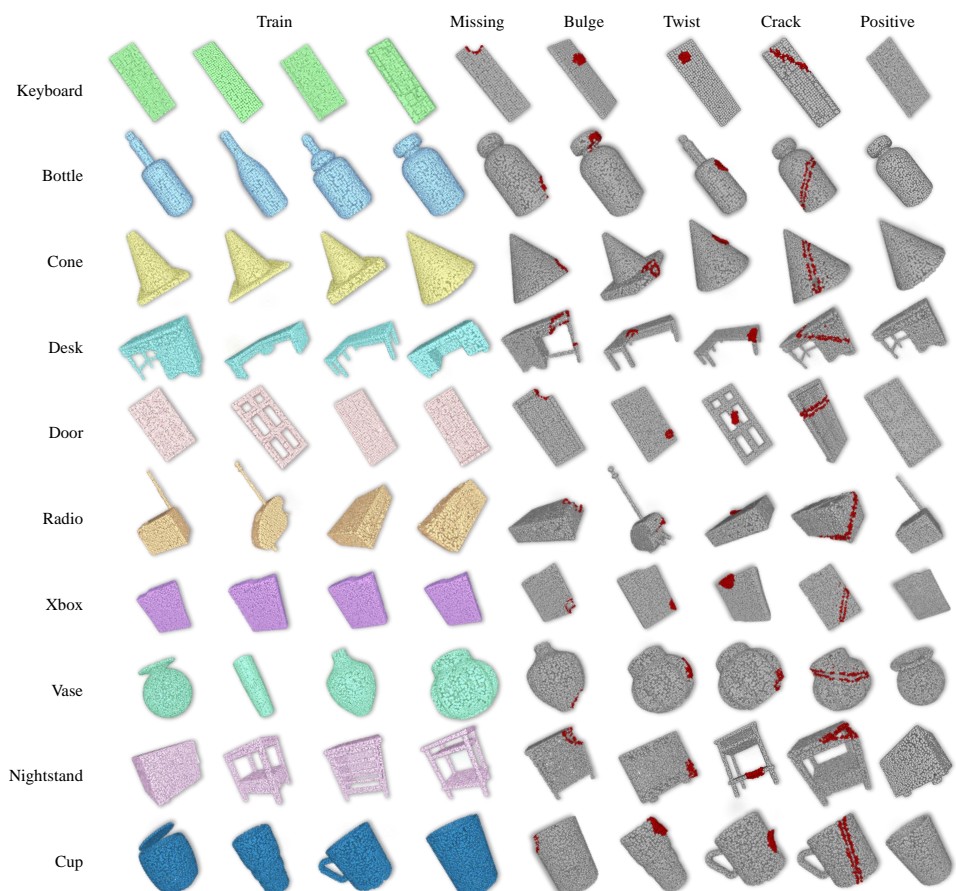

Figure 3: **Visualization of ICD datasets**. We show a training set for some of the classes, along with a sample for each exception. The training set for each class consists of four objects of the same species with different morphologies and produces associated exceptions.

## A.2 DATA PRODUCTION

We utilized several classes from the ModelNet40 dataset as our data source. Since the point clouds in ModelNet40 contain 10,000 points, we upsampled them to 45K–55K points to meet the requirements for anomaly detection Sun et al. (2022). To generate normal samples, we selected four samples from the chosen classes to represent the normal category, applying random rotations and upsampling. Additionally, size variations were introduced through random scaling. Using same-species samples from each subspecies, we generated anomalies based on rotations and scale stretching. We first generated four types of anomalies: local twisting, dents, protrusions, and missing parts. Next, we applied Moving Least Squares (MLS) for initial processing on the generated anomalous point clouds Khabibulin (2023). Finally, we used CloudCompare to refine the anomalous point clouds, obtaining the final samples.

## A.3 DATASET STATISTICS

The dataset consists of four subspecies, with the test set containing between 41 and 64 samples per class, covering both normal and anomalous variations of each subspecies. Each point cloud comprises 45K–55K points, offering a detailed and high-resolution representation of the 3D objects. In comparison to existing 3D anomaly detection datasets, as highlighted in Table 4, our dataset is the first to introduce multi-subspecies anomaly detection. This novel addition not only broadens the scope of anomaly detection tasks by incorporating more granular variations within classes, but also significantly increases the complexity of the detection challenge. As a result, it provides a more

| P-AUROC | | | | | | | | | | | | | |
| Method | cap0 | cap3 | helmet3 | cup0 | bowl4 | vase3 | headset1 | eraser0 | vase8 | cap4 | vase2 | vase4 | helmet0 | bucket1 |
|---|---|---|---|---|---|---|---|---|---|---|---|---|---|---|
| BTF (Raw) | 0.668 | 0.527 | 0.526 | 0.403 | 0.664 | 0.717 | 0.515 | 0.525 | 0.424 | 0.648 | 0.410 | 0.425 | 0.553 | 0.321 |
| BTF (FPFH) | 0.618 | 0.522 | 0.444 | 0.586 | 0.609 | 0.699 | 0.490 | 0.719 | 0.668 | 0.520 | 0.546 | 0.510 | 0.571 | 0.633 |
| M3DM | 0.557 | 0.423 | 0.374 | 0.539 | 0.464 | 0.439 | 0.617 | 0.627 | 0.663 | 0.777 | 0.737 | 0.476 | 0.526 | 0.501 |
| Patchcore (FPFH) | 0.580 | 0.453 | 0.404 | 0.600 | 0.494 | 0.449 | 0.637 | 0.657 | 0.662 | 0.757 | 0.721 | 0.506 | 0.546 | 0.551 |
| Patchcore (PointMAE) | 0.589 | 0.476 | 0.424 | 0.610 | 0.501 | 0.460 | 0.627 | 0.677 | 0.663 | 0.727 | 0.741 | 0.516 | 0.556 | 0.561 |
| CPMF | 0.601 | 0.551 | 0.420 | 0.497 | 0.683 | 0.582 | 0.458 | 0.689 | 0.529 | 0.553 | 0.582 | 0.514 | 0.555 | 0.601 |
| RegAD | 0.693 | 0.725 | 0.367 | 0.510 | 0.663 | 0.650 | 0.610 | 0.343 | 0.620 | 0.643 | 0.605 | 0.500 | 0.600 | 0.752 |
| IMRNet | 0.737 | 0.775 | 0.573 | 0.643 | 0.676 | 0.700 | 0.676 | 0.676 | 0.548 | 0.652 | 0.614 | 0.524 | 0.597 | 0.771 |
| R3D-AD | 0.822 | 0.730 | 0.707 | 0.822 | 0.744 | 0.742 | 0.795 | 0.890 | 0.721 | 0.681 | 0.752 | 0.630 | 0.757 | 0.756 |
| Ours | 0.933 | 0.846 | 0.558 | 1.000 | 0.974 | 0.833 | 0.733 | 0.948 | 0.939 | 0.749 | 0.824 | 0.615 | 0.716 | 0.660 |

| Method | bottle3 | vase0 | bottle0 | tap1 | bowl0 | bucket0 | vase5 | vase1 | vase9 | ashtray0 | bottle1 | tap0 | phone | cup1 |
|---|---|---|---|---|---|---|---|---|---|---|---|---|---|---|
| BTF (Raw) | 0.568 | 0.531 | 0.597 | 0.573 | 0.564 | 0.617 | 0.585 | 0.549 | 0.564 | 0.578 | 0.510 | 0.525 | 0.563 | 0.521 |
| BTF (FPFH) | 0.322 | 0.342 | 0.344 | 0.546 | 0.509 | 0.401 | 0.409 | 0.219 | 0.268 | 0.420 | 0.546 | 0.560 | 0.571 | 0.610 |
| M3DM | 0.510 | 0.423 | 0.574 | 0.739 | 0.634 | 0.309 | 0.317 | 0.427 | 0.663 | 0.577 | 0.637 | 0.754 | 0.357 | 0.556 |
| Patchcore (FPFH) | 0.572 | 0.455 | 0.604 | 0.766 | 0.504 | 0.469 | 0.417 | 0.423 | 0.660 | 0.587 | 0.667 | 0.753 | 0.388 | 0.586 |
| Patchcore (PointMAE) | 0.650 | 0.447 | 0.513 | 0.538 | 0.523 | 0.593 | 0.579 | 0.552 | 0.629 | 0.591 | 0.601 | 0.458 | 0.488 | 0.556 |
| CPMF | 0.405 | 0.451 | 0.520 | 0.697 | 0.783 | 0.482 | 0.618 | 0.345 | 0.609 | 0.353 | 0.482 | 0.359 | 0.509 | 0.499 |
| RegAD | 0.525 | 0.533 | 0.486 | 0.641 | 0.671 | 0.610 | 0.520 | 0.702 | 0.594 | 0.597 | 0.695 | 0.676 | 0.414 | 0.538 |
| IMRNet | 0.640 | 0.533 | 0.552 | 0.696 | 0.681 | 0.580 | 0.676 | 0.757 | 0.594 | 0.671 | 0.700 | 0.676 | 0.755 | 0.757 |
| R3D-AD | 0.781 | 0.788 | 0.733 | 0.900 | 0.819 | 0.683 | 0.757 | 0.729 | 0.718 | 0.833 | 0.737 | 0.736 | 0.762 | 0.757 |
| Ours | 0.991 | 0.829 | 0.900 | 0.633 | 0.978 | 0.921 | 0.615 | 0.791 | 0.647 | 0.891 | 0.933 | 0.735 | 0.995 | 0.702 |

| Method | vase7 | helmet2 | cap5 | shelf0 | bowl5 | bowl3 | helmet1 | bowl11 | headset0 | bag0 | bowl2 | jar | Mean |
|---|---|---|---|---|---|---|---|---|---|---|---|---|---|
| BTF (Raw) | 0.448 | 0.602 | 0.373 | 0.164 | 0.417 | 0.385 | 0.349 | 0.264 | 0.378 | 0.410 | 0.525 | 0.420 | 0.493 |
| BTF (FPFH) | 0.518 | 0.542 | 0.586 | 0.609 | 0.699 | 0.490 | 0.719 | 0.668 | 0.520 | 0.546 | 0.510 | 0.424 | 0.528 |
| M3DM | 0.657 | 0.623 | 0.639 | 0.564 | 0.409 | 0.617 | 0.427 | 0.663 | 0.577 | 0.537 | 0.684 | 0.441 | 0.552 |
| Patchcore (FPFH) | 0.693 | 0.425 | 0.790 | 0.494 | 0.558 | 0.537 | 0.484 | 0.639 | 0.583 | 0.571 | 0.615 | 0.472 | 0.568 |
| Patchcore (PointMAE) | 0.650 | 0.447 | 0.538 | 0.523 | 0.593 | 0.579 | 0.552 | 0.629 | 0.591 | 0.601 | 0.458 | 0.483 | 0.562 |
| CPMF | 0.397 | 0.462 | 0.697 | 0.685 | 0.685 | 0.658 | 0.589 | 0.639 | 0.643 | 0.643 | 0.625 | 0.610 | 0.559 |
| RegAD | 0.462 | 0.614 | 0.467 | 0.688 | 0.593 | 0.348 | 0.381 | 0.525 | 0.537 | 0.706 | 0.490 | 0.592 | 0.572 |
| IMRNet | 0.635 | 0.641 | 0.652 | 0.603 | 0.710 | 0.599 | 0.600 | 0.702 | 0.720 | 0.660 | 0.685 | 0.780 | 0.661 |
| R3D-AD | 0.771 | 0.633 | 0.670 | 0.696 | 0.656 | 0.767 | 0.720 | 0.778 | 0.738 | 0.720 | 0.741 | 0.838 | 0.749 |
| Ours | 0.762 | 0.887 | 0.733 | 0.852 | 0.519 | 0.863 | 0.648 | 0.904 | 0.891 | 0.705 | 1.000 | 0.957 | 0.815 |

Table 5: The experimental results P-AUROC (↑) for anomaly detection across 40 categories of Anomaly-ShapeNet. The best and the second-best results are highlighted in **red** and **blue**, respectively.

| P-AUROC | | | | | | | | | | | |
| Method | bottle | cup | desk | door | keyboard | night_stand | radio | vase | xbox | cone | Mean |
|---|---|---|---|---|---|---|---|---|---|---|---|
| BTF (Raw) | 0.377 | 0.375 | 0.305 | 0.234 | 0.314 | 0.363 | 0.448 | 0.413 | 0.285 | 0.347 | 0.346 |
| BTF (FPFH) | 0.579 | 0.557 | 0.452 | 0.515 | 0.424 | 0.424 | 0.528 | 0.588 | 0.515 | 0.556 | 0.514 |
| M3DM | 0.437 | 0.422 | 0.361 | 0.308 | 0.336 | 0.336 | 0.421 | 0.478 | 0.361 | 0.444 | 0.391 |
| Patchcore (FPFH) | 0.684 | 0.581 | 0.504 | 0.521 | 0.474 | 0.549 | 0.653 | 0.569 | 0.695 | 0.583 | 0.581 |
| Patchcore (PointMAE) | 0.568 | 0.618 | 0.482 | 0.263 | 0.395 | 0.517 | 0.520 | 0.577 | 0.537 | 0.574 | 0.505 |
| RegAD | 0.362 | 0.589 | 0.321 | 0.261 | 0.387 | 0.477 | 0.314 | 0.573 | 0.487 | 0.554 | 0.333 |
| Ours | 0.686 | 0.573 | 0.490 | 0.542 | 0.467 | 0.547 | 0.638 | 0.541 | 0.682 | 0.578 | 0.574 |

Table 6: The experimental results I-AUROC (↑) for anomaly detection across 10 categories of ICD. The best and the second-best results are highlighted in **red** and **blue**, respectively.

rigorous and comprehensive evaluation platform for testing the effectiveness and generalizability of anomaly detection models across diverse 3D geometries and subspecies configurations.

# B   MORE EXPERIMENT RESULT

We present additional IGB-AD results for the ANOMALA-SHAPENet and ICD datasets in Table 2 and Table 6, respectively. Experiments show that our IGB-AD method has a wide range of advantages in terms of generalization and robustness.

# C   FURTHER UNDERSTANDING OF IGB

To help better understand IGB, we use noise characterization to visualize the IGB process in Figure 4. It starts with complex noise, containing useful and useless information. With each processing, useless information in the noise is continuously removed and useful information is eventually recorded. The useful information is the extended information based on the original feature matrix in anomaly detection, which helps to distinguish the normal and abnormal features in anomaly detection.

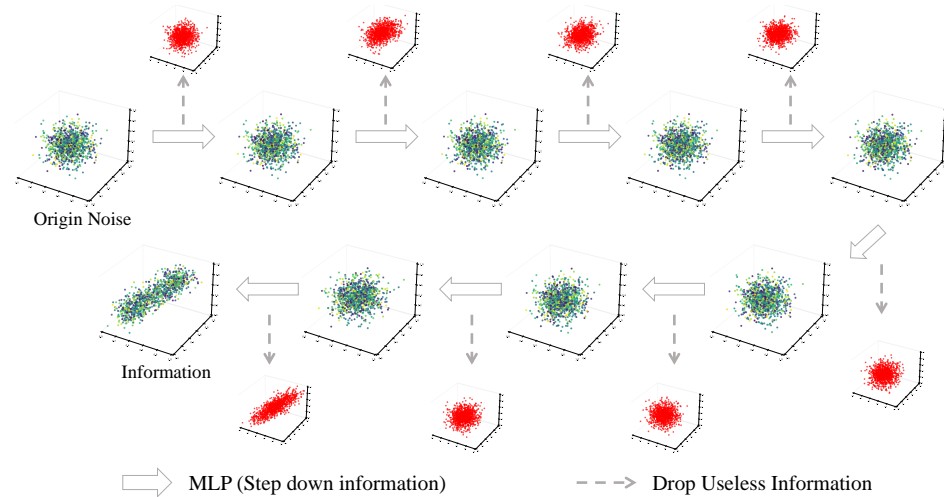

Figure 4: **Qualitative diagram of the process of IGB layer operation.** We simulated the workflow of IGB module with a Gaussian noise, and the useless was removed continuously, and finally the useful information was left.

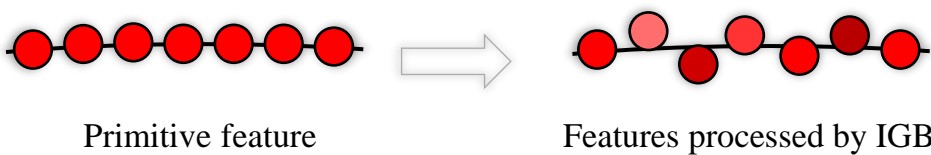

Primitive feature             Features processed by IGB

Figure 5: **The features before and after IGB processing were compared.** We use a curve to describe the mean of features, similar structures have similar features, distributed around the mean curve.

We visualized before and after features were processed by IGB in Figure 5. On the left, we show that under similar structures, the extracted similar features are closely clustered around the mean curve, making them difficult to differentiate in anomaly detection. After processing with IGB, we obtain the feature distribution displayed on the right, where the differentiated feature representations are more conducive to anomaly detection.

# D    PSEUDOCODE

We then provide pseudo-code for PD, RIFPS, IGB, and IP in turn

---

**Algorithm 1** Packet Downsampling (PD)

---

**Input:** Feature matrix $F$, number of clusters $C$, downsampling ratio $r$
**Output:** Downsampled feature matrix $F_{\text{down}}$
  1: **Step 1: Group features into $C$ clusters**
  2: Apply clustering (e.g., k-means) on $F$ to form $C$ clusters: $\{C_1, C_2, \ldots, C_C\}$
  3: **Step 2: Select representative feature from each cluster**
  4: **for** each cluster $C_j$ **do**
  5:     Compute importance score $S(f_i)$ for each feature $f_i \in C_j$
  6:     Select the feature $f_{\text{rep}}$ with the highest score in cluster $C_j$
  7:     Add $f_{\text{rep}}$ to the downsampled set $F_{\text{down}}$
  8: **end for**
  9: **Step 3: Return the downsampled feature matrix** $F_{\text{down}}$
10: **return** $F_{\text{down}}$

---

---

**Algorithm 2** Rotation-Invariant Farthest Point Sampling (RIFPS)

---

**Input:** Point cloud $P = \{p_1, p_2, \ldots, p_N\}$, number of points $N$, desired sample size $M$
**Output:** Sampled point set $F$
 1: **Step 1: Compute the geometric center** $C$ of the point cloud
 2: $C \leftarrow \frac{1}{N} \sum_{i=1}^{N} p_i$
 3: **Step 2: Calculate the Euclidean distance** from each point $p_i$ to the center $C$
 4: **for** $i = 1$ to $N$ **do**
 5:     $D_i \leftarrow \|p_i - C\|_2$
 6: **end for**
 7: **Step 3: Select the farthest point** $p_f$ from the center
 8: $p_f \leftarrow \arg\max_i D_i$
 9: **Step 4: Perform iterative Farthest Point Sampling (FPS)**
10: Initialize the sampled set $F$ with the farthest point $p_f$
11: $F \leftarrow \{p_f\}$
12: **for** $j = 2$ to $M$ **do**
13:     Find the point $p_{\text{next}}$ that maximizes the minimum distance to any point in $F$
14:     $p_{\text{next}} \leftarrow \arg\max_{p \in P} \min_{f \in F} \|p - f\|_2$
15:     Add $p_{\text{next}}$ to the sampled set $F$
16:     $F \leftarrow F \cup \{p_{\text{next}}\}$
17: **end for**
18: **Return** the sampled point set $F$

---

**Algorithm 3** Information Gain Block (IGB)

---

**Input:** Noise matrix $Z$, feature matrix $F^{(i-1)}$
**Output:** Extracted useful information $X$
 1: **Step 1: Decompose noise matrix** $Z$
 2: Decompose $Z$ into useful information $X$ and irrelevant noise $Y$:
 3: $Z \leftarrow X + Y$
 4: Model $Y$ as Gaussian noise: $Y \sim \mathcal{N}(0, \sigma^2)$
 5: **Step 2: Extract useful information** $X$ **using MLP**
 6: $X \leftarrow f_{\text{MLP}}(Z, F^{(i-1)})$
 7: **Step 3: Apply loss functions to ensure relevance of extracted information**
 8: Compute smoothness loss:
 9: $L_{\text{smooth}} \leftarrow \text{SmoothL1Loss}(F^{(i-1)}, F^{(i-1)} + X)$
10: Compute richness loss:
11: $L_{\text{richness}} \leftarrow \text{sigmoid}(\alpha \cdot \text{mean}(\text{Var}(F^{(i-1)} + X)))$
12: Total loss:
13: $L_{\text{total}} \leftarrow \beta \cdot L_{\text{smooth}} + \lambda \cdot (1 - L_{\text{richness}})$
14: **Step 4: Return the extracted useful information**
15: **return** $X$

---

**Algorithm 4** Information Perfusion (IP) with Stacked IGBs

---

**Input:** Initial feature matrix $F^{(0)}$, noise matrix $Z$, number of layers $k$
**Output:** Enhanced feature matrix $F^{(k)}$
 1: **Step 1: Initialize feature matrix** $F^{(0)} \leftarrow F^{\text{initial}}$
 2: **Step 2: Perform stacked IGBs for** $k$ **layers**
 3: **for** $i = 1$ to $k$ **do**
 4:     Apply Information Gain Block (IGB) on $F^{(i-1)}$
 5:     $X^{(i)} \leftarrow \text{IGB}(Z, F^{(i-1)})$
 6:     $F^{(i)} \leftarrow F^{(i-1)} + X^{(i)}$
 7: **end for**
 8: **Step 3: Return the enhanced feature matrix** $F^{(k)}$
 9: **return** $F^{(k)}$

---

