# OpenReview forum: "Noise is More Than Just Interference: Information Infusion Networks for Anomaly Detection"
_ICLR.cc/2025/Conference — ICLR 2025 Conference Withdrawn Submission_

### Official Review · Reviewer_2S64 · 2024-10-24

**Soundness:** 2
**Presentation:** 2
**Contribution:** 2
**Rating:** 3
**Confidence:** 5

**Summary:**

This paper proposes Information Gain Block-based Anomaly Detection (IGB-AD) for 3D anomaly detection to address the challenges of insufficient anomaly detection information and high intra-class variance. Overall, the writing is not clear, and the experimental results fail to demonstrate the superiority of the proposed method.

**Strengths:**

This paper offers a comprehensive literature review.

**Weaknesses:**

it would be better to specify your title to include 3D anomaly detection or point cloud anomaly detection to be more specific.

What is your definition of noise? The lack of definition makes the motivation hard for me to understand. like, "Noise, as a prior source of information, consists of a combination of various types of data", what is noise?

Why the teacher-student distillation networks are proposed to mitigate the effects of noise in Lines 65 and 67? I am not convinced by this claim. Like in 3DST, RD4AD, CDO, etc., is there any technique related to noise?

the description of the method is hard to understand as well. It would be better if you could improve the overview of your method a bit. Currently, I am not clear about your motivation for the framework, yet the relationships between the proposed components and the motivation are unclear.

The authors only conduct experiments on Anomaly-Shapenet and the established dataset. What about Real3D and MVTec 3D?

We can see in Table 1, that the proposed method can even perform worse than a simple baseline FPFH in some categories, which is confusing and fails to demonstrate the effectiveness of the proposed method.

Also, what about the point-level results? In Table 1 and Table 2, only object-level results are presented.

The ablation results in Table 3 fail to demonstrate the effectiveness of individual components since the variation is not significant enough. We can see that with only PD, the authors even achieve higher P-AUROC than some other variants like in rows 1, and 4 of Table 3.

**Questions:**

See the weakness.

---

### Official Review · Reviewer_bPxg · 2024-10-26

**Soundness:** 2
**Presentation:** 1
**Contribution:** 2
**Rating:** 5
**Confidence:** 4

**Summary:**

This manuscript claims that most existing 3D anomaly detection methods require the usage of registration to preprocess point clouds and exhibit high intra-class variance. To this end, it proposes IGB, IGB-AD, RIFPS, IP, and PD module to enhance 3D anomaly detection and alleviate these two challenges. Furthermore, it develops an Intra-Class Diversity (ICD) 3D dataset with multiple subclasses. Moreover, the proposed method achieves the state-of-the-arts performance on one public dataset and the proposed dataset.

**Strengths:**

1.This manuscript proposes IGB, IGB-AD, RIFPS, IP, and PD module to enhance 3D anomaly detection.

2.This manuscript introduces the ICD dataset for 3D anomaly detection. Different from existing datasets, it includes multiple sub-classes.

3.The method proposed in the manuscript achieves the state-of-the-arts performance on one public dataset and the proposed dataset

**Weaknesses:**

1.The citation format is incorrect, with many references needing to be placed in parentheses. The authors should carefully read Section 4.1 of the Formatting Instructions for ICLR 2025 Conference Submissions.

2.Lack of details on ICD datasets.  Since the ICD dataset is the second contribution, the motivation for its creation should be described in the introduction section.

3.The Introduction section could be better articulated. The author spends most of the Introduction describing the current issues with 3D anomaly detection but does not explain how their proposed method effectively addresses these challenges. Deeper insights need to be provided.

4.In Page-2 Line-68, R3D-AD reconstructs normal samples from pseudo abnormal point clouds using a Diffusion model and cannot be categorized as a distillation method.

5.Lack of experiments. 1) the ablation and comparison experiment on the proposed Rotation-Invariant Farthest Point Sampling (RFPS)   2)The performance of the proposed method on the Real3D-AD dataset.

**Questions:**

1. The definition of 'prior noise' is missing. The authors mention 'prior noise' in the abstract and introduction but do not provide a definition, nor is it described in the methods section.

2. How does the proposed method tackle the challenge of high intra-class variance?

3. In Page-2 Line-77, what is the link between extracting valuable information and high intra-class variance?

4. In Table 3, the unit for "Time Cost" needs to be provided, whether it is seconds or milliseconds.

5.  Are there any hyperparameters in the proposed method? Are they sensitive?

---

### Official Review · Reviewer_6AHZ · 2024-11-01

**Soundness:** 2
**Presentation:** 2
**Contribution:** 2
**Rating:** 5
**Confidence:** 5

**Summary:**

The authors propose an Information Gain Block-based Anomaly Detection method to address the issue of high intra-class variance. They introduce Rotation-Invariant Farthest Point Sampling and an Information Perfusion module composed of Information Gain Blocks. The authors incorporate noise into 3D anomaly detection to provide more distinctive feature information. Additionally, they construct the Intra-Class Diversity (ICD) 3D anomaly detection dataset. The effectiveness of the method is validated on the constructed dataset and the ShapeNet dataset.

**Strengths:**

The authors propose an Information Gain Block-based Anomaly Detection method to address the issue of high intra-class variance. They introduce Rotation-Invariant Farthest Point Sampling and an Information Perfusion module composed of Information Gain Blocks. The authors incorporate noise into 3D anomaly detection to provide more distinctive feature information. Additionally, they construct the Intra-Class Diversity (ICD) 3D anomaly detection dataset

**Weaknesses:**

See questions.

**Questions:**

1. The authors seem to achieve better performance by stacking layers of IGB and increasing the number of MLPs within them. Is this performance improvement due to increased computational complexity?
2. In Table 3, the results without using IP and IGB appear to be better than those with IP and two layers of IGB. Please explain the effectiveness of IGB and IP.
3. The comparison methods in Table 1 differ from those in Table 2. It seems that the experimental results of CPMF, IMRNet, and R3D-AD on the ICD dataset are missing in Table 2. It is recommended that the authors include these results to demonstrate the reliability of the experiments.
4. The proposed dataset does not seem to have a significant advantage in terms of defect types and quantity. It appears to be a selection of a few subclasses from each category in the ModelNet dataset.

---

### Official Review · Reviewer_doed · 2024-11-02

**Soundness:** 1
**Presentation:** 3
**Contribution:** 3
**Rating:** 5
**Confidence:** 4

**Summary:**

The authors propose a novel method that uses a noise prior to learn to improve the features of a handcrafted descriptor FPFH. The FPFH features are reformulated through a series of Information Gain blocks that attempt to extract useful information from noised FPFH features thus decoupling the noise from the useful information contained within the features. The extracted features are then used to create a memory bank which is used at inference for anomaly score estimation. A packet downsampling process is also proposed, which is a Mahalanobis distance-based greedy coreset sampling mechanism that better samples features in cases where the observed class is composed of several subclasses.

The authors also propose a new dataset, ICD, where each class is composed of several subclasses providing a unique challenge for 3D anomaly detection methods.

**Strengths:**

- Interesting method that aims to improve existing handcrafted features which seems novel.
- The proposed method achieves state-of-the-art results on the AnomalyShapeNet dataset and on the newly proposed ICD dataset.
- Most sections of the paper are well written and easy to follow despite the proposed method being constructed of several components.

**Weaknesses:**

Some implementation details, such as how noise is injected and sampled during the pretraining phase, could be included as it would improve clarity for the reader. Some implementation details are included in the supplementary but could be moved to the main paper.

The results of the comparison of the proposed method to related works could be discussed in more detail. Currently the results on AnomalyShapeNet and ICD are only briefly listed in Section 4.3, however no discussion of the results is given.

On the ICD dataset the current SOTA on AnomalyShapeNet is not evaluated (R3D-AD) and the second best method is a vanilla PatchCore using FPFH features which generally does not achieve SOTA results on 3D anomaly detection benchmarks. Given that the ICD dataset is one of the claimed contributions of this paper the evaluation should be more thorough and the discussion of the results more detailed.

The ablation study is done on the newly proposed ICD where the performance is very low (0.6 AUC). This makes it difficult to really evaluate the components of the method since most anomalies are already missed and the difference between most experiments is less than 1% AUROC.

Overall I believe the experimental section is the most lacking. There is a lack of discussion of the results on both the AnomalyShapeNet and the ICD dataset. Additionally, the evaluation on the ICD dataset could be more thorough. Methods that are included in the AnomalyShapeNet experiments are not included in the ICD experiments. The results are not properly discussed. Only  image-level AUROC is used for the evaluation in Section 4.3 but in the ablation study (Section 4.4) other metrics are also used. The ablation study should also be done on AnomalyShapeNet to get a clearer picture of the impact of each design choice.

**Questions:**

In Eq. 7, L_richness. It would be useful to give dimensions of X and F. Possibly also to rewrite the equation to make the way this is calculated easier to understand.

In Sec 3.3. - How is noise Z added? Is it sampled once and used for all blocks?

In Eq. 11, which features are max(s) and min(s) calculated from in the normalization?

Why no M3DM comparison or comparison on MVTec3D or on Real3DAD that have been published and are more widely cited?

The BTF method achieves an extremely low AUROC score on the ICD dataset showing a strong correlation between the anomaly score and the normality of the example which may be interesting and should be commented on given that the dataset is one of the contributions.

The discussion of the experimetnal results could be expanded.

Why are the results on the ICD dataset relatively low in terms of the AUROC scores?

---

### Note · Authors · 2024-11-15

I have read and agree with the venue's withdrawal policy on behalf of myself and my co-authors.